# Development and Evaluation of a Gatekeeper Training Program Regarding Anxiety about Radiation Health Effects Following a Nuclear Power Plant Accident: A Single-Arm Intervention Pilot Trial

**DOI:** 10.3390/ijerph17124594

**Published:** 2020-06-26

**Authors:** Masatsugu Orui, Maiko Fukasawa, Naoko Horikoshi, Yuriko Suzuki, Norito Kawakami

**Affiliations:** 1Department of Public Health, School of Medicine, Fukushima Medical University, Fukushima 960-1295, Japan; copepe@fmu.ac.jp (N.H.); yrsuzuki@gmail.com (Y.S.); 2Sendai City Mental Health and Welfare Center, Sendai 980-0845, Japan; 3Department of Mental Health, Graduate School of Medicine, The University of Tokyo, Tokyo 113-0033, Japan; fukasawa@m.u-tokyo.ac.jp (M.F.); norito@m.u-tokyo.ac.jp (N.K.); 4Radiation Medical Science Center for the Fukushima Health Management Survey, Fukushima Medical University, Fukushima 960-1295, Japan; 5National Institute of Mental Health, National Center of Neurology and Psychiatry, Tokyo 187-8553, Japan

**Keywords:** gatekeeper, mental health first aid, nuclear disaster, radiation, anxiety

## Abstract

After the Fukushima Daiichi Nuclear Power Plant accident in March 2011, residents perceived a radiation exposure risk. To address the concerns about radiation exposure and the subsequent effects on their health, we developed the gatekeeper training program for radiation health anxiety and mental health issues. The program consisted of a presentation and roleplay, with educational objectives to the increase knowledge and understanding around radiation health anxiety, alcoholism, depression, and suicide. Twenty-six counselors participated in the program as a single-arm intervention. To measure the outcomes, the subjects reported self-confidence when they counselled with residents with radiation health anxiety and other mental health issues. Comparing the answers to pre- and 2-month follow-up surveys, the confidence levels were higher after 2 months than at baseline, especially for the question “How confident are you at supporting a resident with mental health issues?”, which showed the largest increase (*p* < 0.001). Regarding radiation health anxiety the questions “Can you communicate coping skills to a resident with radiation health anxiety?” (*p* = 0.007) and “Can you refer a resident with radiation health anxiety to professionals who can cope appropriately?” (*p* = 0.016) showed significant increases in their confidence levels. This program could be useful for on-going health activities and future nuclear disasters.

## 1. Introduction

The Great East Japan Earthquake (GEJE), which occurred on March 11 2011, was the largest earthquake ever recorded in Japan’s history and consequently generated a devastating tsunami. This was followed by a separate tsunami, which hit the Fukushima Daiichi Nuclear Power Plant operated by the Tokyo Electric Power Company (TEPCO), causing a radiation disaster in the Fukushima prefecture, which required the long-term evacuation of residents from many of the surrounding municipalities. Due to this nuclear accident, the Japanese government created designated evacuation areas, resulting in more than 164,000 residents who lived near the nuclear power plant being forced to leave their homes in May 2012 [1]. The evacuation orders from the Japanese government have been gradually removed since 2015 in 10 of the 12 municipalities designated as evacuation zones (by April 2017).

Radiation exposure from the nuclear disaster was an unprecedented experience for the residents, and their perceived radiation exposure risk may have affected their disaster-related stress or psychological distress [2,3,4]. One year after the disaster (February 2012), 47.6% of residents who had lived in the evacuation areas in Fukushima perceived that they were “likely” or “very likely” to have delayed health effects (e.g., cancer onset) as a result of their current level of radiation exposure [5]. Even when the evacuation order had already been lifted, and some residents had returned to their homes, the proportion of those who perceived that they were “likely” or “very likely” to have delayed radiation health effects was 34.0%, even 7 years after the GEJE [6,7]. Moreover, residents in the Fukushima evacuation area not only have concerns about exposure to radiation and the effects on their health, but they also have other mental health problems (e.g., overdrinking alcohol or alcoholism, depression, and suicide) related to the nuclear disaster and subsequent changes in their personal lives and environment due to the evacuation [8,9,10,11,12,13].

Following the GEJE, in order to support evacuees living in temporary housing or new, permanent living arrangements and also to cope with evacuees’ mental health issues, as described above (e.g., alcoholism, depression and suicide), a large number of livelihood support counselors were employed by the Social Welfare Councils in the Fukushima, Miyagi, and Iwate prefectures. The counselors’ tasks are 1) to carry out regular visits to evacuees to support them after the evacuation, 2) to listen carefully and non-judgmentally to reduce their feelings of unease, 3) to connect with mental health and other professionals, and 4) to coordinate a “watch-over system” at a community level [14]. A systematic review by Mann et al. included education and awareness for the general public and professionals (e.g., gatekeeper training) as one of five recommended suicide prevention strategies [15]. Gatekeeper training is often integrated into suicide prevention strategies aimed at educating social and community facilitators to identify signs of suicidal behavior and to signpost individuals to appropriate services [15]. Although few studies have reported the gatekeeper program as being effective for reducing suicide rates at a community level, there are reports that community-based multimodal interventions for suicide prevention, including the use of gatekeepers, have contributed to lower suicide rates in Japanese rural areas [16]. Therefore, we hypothesized that livelihood support counselors who support local residents would play the role of gatekeepers, which could lead to a reduction in radiation health anxiety and other mental health issues among residents in Fukushima through the gatekeeper training programs.

However, few gatekeeper training programs are focused on post-nuclear disaster and radiation health anxiety, including those used in the event of natural disasters [17]. The Japanese national suicide prevention policy increased the implementation of gatekeeper training programs after the release of “the Basic Act for Suicide Prevention” and “General Principles of Suicide Prevention Policy” by the Japanese Cabinet Office in 2006–2007 [18]. After implementing the “Basic Act for Suicide Prevention,” several gatekeeper programs in Japan (targeting various subjects such as medical staff, general office workers, or administration staff in universities) [19,20,21,22,23] have been evaluated, but there remains a need for the integration of the development and evaluation of gatekeeper training programs in post-disaster settings [23].

Therefore, focusing on the role of gatekeepers in offering appropriate services to general residents, we developed a gatekeeper training program for evacuees and residents in Fukushima who hold negative perceptions of radiation health effects with the aim of providing appropriate services to reduce their health concerns or anxiety and improve their mental health status. In addition, we evaluated the participants’ self-confidence in participating in counseling and responding to residents with radiation health anxiety and mental health issues through the developed program (see Appendix A for entire training program for radiation health anxiety and other mental health issues).

## 2. Materials and Methods

### 2.1. Survey of the Gatekeeper Training Program Needs Among Livelihood Support Counselors

Firstly, in conducting a survey of the gatekeeper training program needs and developing the program, we referred to the Mental Health First Aid program (MHFA), which has already been developed and proven to be effective for improving knowledge, reducing stigmatizing attitudes, and increasing supportive behaviors by pre-post studies, controlled trials, and a systematic review [24,25,26,27].

A self-administered questionnaire survey was conducted with livelihood support counselors in the Social Welfare Councils of Minami-Soma City and Iitate Village, which were areas in which the evacuation order had already been lifted. The Fukushima City Social Welfare Council also conducted a survey to represent the non-evacuation area as a control (Figure 1). This questionnaire survey was distributed using a placement method for counselors of three Social Welfare Councils in November 2018. This study targeted all counselors in the three Social Welfare Councils. Therefore, the total number of subjects in this survey was 44 counselors among the Minami-Soma City (*n* = 20), Iitate Village (*n* = 13) and Fukushima City (*n* = 11) Social Welfare Councils. In order to understand the difference in coping ability of residents with radiation health anxiety, we targeted Social Welfare Councils located in both evacuation and non-evacuation areas.

The survey contents consisted of age, gender, length of service as a counselor, certificated health or welfare license holder or not, history of coping of residents with any degree of radiation health anxiety and other mental health issues (depression, suicidal thoughts, schizophrenia, alcoholism, dementia, personality disorders, and developmental disorders). The reason why we asked about these mental health issues was that the Mental Health First Aid (MHFA) program [24,25,26,27] deals with these five mental health modules (depression, suicidal thoughts, schizophrenia, and alcoholism). Besides, counselors are likely to address residents who have dementia, personality disorders, and developmental disorders. Therefore, we investigated the coping history of residents with these eight mental health issues. Finally, we asked about their needs for the training program (e.g., how to deal with residents with mental health issues including radiation health anxiety) by using a column for further comments in the questionnaire.

For the data analysis of the further comments in the questionnaire, 1) radiation health anxiety, 2) other mental health issues, and 3) their needs for the training program were analyzed by text mining while searching the frequency of words and reviewing sentences containing frequently used words.

### 2.2. Development of the Gatekeeper Training Program for Radiation Health Anxiety and Mental Health Issues

Based on the results of experiences of coping with residents with any radiation health anxiety or other mental health issues and their needs for the gatekeeper training program, we developed a program for radiation health anxiety and other mental health issues that incorporates the basic elements of the Mental Health First Aid program (MHFA) [24,25,26,27]. The MHFA program is an officially qualified first-aid course for non-health professionals that aims to improve mental health literacy [24,25]. The MHFA-Japan team was established in 2007, and their team cooperates with the gatekeeper training program of the cabinet office of the Japanese government [28].

### 2.3. Evaluation of the Gatekeeper Training Program

In November and December 2019, 26 livelihood support counselors from Iitate Village and Minami-Soma City Social Welfare Council participated in the training program; this intervention study was designed as a single-arm trial study, but with a lack of a control group and insufficient sample size. Self-administered questionnaires were distributed to the study participants in order to evaluate the outcomes of the training program, including i) a pre-intervention survey, ii) an immediate post-intervention survey, and iii) a follow-up survey 2 months after the intervention was conducted. These three surveys measured the participants’ self-confidence in counseling and responding to residents with radiation health anxiety and mental health issues using a 5-degree scale (1: “not confident,” 2: “not very confident,” 3: “neither confident or not confident,” 4: “a little confident,” and 5: “confident”). The survey measurements were investigator-designed.

We compared the scores of the pre-intervention, immediate post-intervention, and 2-month follow-up survey using a paired t-test with a 5% level of significance. All statistical analyses were performed using SPSS 25.0 (IBM Corp., Armonk, NY, USA).

### 2.4. Ethical Considerations

The survey was approved by the Ethical Review Committee of Fukushima Medical University on 29 October, 2018 (No. 30129) for the survey of gatekeeper training program needs among livelihood support counselors and on 16 July, 2019 (No. 2019-111) for the evaluation of the training program. The trial was registered as UMIN000038462 by UMIN (University Hospital Medical Information Network) Clinical Trials.

## 3. Results

### 3.1. Survey of the Gatekeeper Training Program Needs Among Livelihood Support Counselors

We conducted a self-administered questionnaire survey of all livelihood support counselors in the Minami-Soma City, Iitate Village, and Fukushima City Social Welfare Councils. Valid responses were obtained from 40 counselors (response rate 90.9%). Those who had worked in the evacuation area (Iitate and Minami-Soma Social Welfare Council) had longer serving periods as counselors and had more experience in dealing with residents with radiation health anxiety and other mental health issues than those who had worked in the non-evacuation area. However, there were few certificated health or welfare license holders in the evacuation area. On the other hand, there were few counselors in the non-evacuation area with experience of working with residents with radiation health anxiety and other mental health issues (Table 1). Additionally, regarding other mental health issues that livelihood support counselors had experience with, dementia (53.3%), depression (50.0%), and alcoholism (40.0%) were prominent among counselors in the evacuation area.

As a result of word analysis by text mining, the word “radiation” was extracted. The counselors were able to refer to other specialized organizations such as “recommending to other facilities to inspect radiation level” and “guiding to a specialized consultation agency.” On the other hand, regarding “radiation” issues and concerns faced by residents, there was concern about the dose of radiation in crops or foods and about the mental burden associated with future life when returning home to where they had lived before evacuation. Regarding counselors’ ability to cope with residents with anxiety about radiation, “listening non-judgmentally” was selected. However, it was shown that counselors were frustrated because they could not give effective advice to residents with concerns about radiation only by listening non-judgmentally. On the other hand, some counselors gave advice about actually measuring air radiation levels or radiation doses of crops as well as listening non-judgmentally.

As for mental health issues other than radiation health anxiety, “coordination and connection to other specialized facilities,” “alcohol or alcoholism,” and “listening non-judgmentally” were extracted. Alcohol addiction was identified as a mental health issue among residents; however, in many cases, the counselors themselves expressed “difficulty coping with a resident with alcohol issues” and that “their coping skills were insufficient.” Therefore, it can be seen that the counselors were faced with a difficult situation when communicating with residents with alcohol issues. Moreover, according to responses from the counselors, the counselors were confused that when supporting a resident who was at a risk of suicide, they could not support them sufficiently but could only listen non-judgmentally. However, some counselors felt that “listening non-judgmentally” with a resident who had a risk of suicide could encourage them to reveal stressed feelings to counselors.

Several opinions were expressed regarding the needs for the gatekeeper training program. Some counselors would like to improve their listening non-judgmentally skills; however, most of them had many opinions on how to cope with residents with mental health issues such as alcoholism or suicidal thoughts and were hoping to obtain more advanced coping skills than simply listening non-judgmentally (see Table 2).

### 3.2. Development of the Gatekeeper Training Program for Radiation Health Anxiety and Mental Health Issues

Considering the results of a survey of training program needs, we developed a gatekeeper training program partly based on Mental Health First Aid (MHFA). The standard MHFA is applicable to diverse conditions including depression, anxiety disorders, psychosis, and substance use disorder. We selected depression, suicidal thoughts, alcoholism, and anxiety (including radiation health anxiety) as mental health modules in this program. The need for training programs to help with alcoholism and suicide among counselors is in high demand. Dementia was the condition with the highest proportion (66.7%) of coping history among counselors; however, residents with dementia are supported by comprehensive general support community centers for the elderly. Therefore, we excluded dementia from the developed program module.

Although the standard MHFA program is a 12-h course, we set this gatekeeper training program as a 100-minute course because 1) 2-h brief suicide intervention programs partially based on the MHFA program have already been shown to be effective [19,20,21,22] and 2) it is a more accessible and less burdensome program when counselors undergo training. To shorten the standard MHFA program, we limited our focus to coping with residents with radiation health anxiety as our main purpose for the intervention and looked at mental health issues (e.g., depression, suicide thoughts, and alcoholism) generated by disaster-related stress and psychological distress [8,9,10,11,12,13]. Therefore, psychosis modules such as schizophrenia were excluded because of the relatively low proportion of counselors with history of coping with residents with these conditions and the lower needs for their inclusion in a training program. Only two vignettes were utilized in the roleplay session to reduce the training time. As a result, the program consisted of two parts in a 100-minute course. We allocated the first 70 min to a presentation (partially including a group discussion) on factual information on anxiety (included radiation health anxiety), alcoholism, depression, and suicidal thoughts. In the second session, roleplay was included because of positive evaluation from participants in gatekeeper training in a previous study [18]. We demonstrated gatekeeper behaviors and had a small group discussion, along with presenting example scenarios, and we improved communication among evacuees through roleplay. In particular, to address radiation health anxiety, we suggested the following to the counselors in the presentation section:
“If a resident was concerned about the air radiation level in their living environment or the radiation dose of crops, it is better to recommend them to measure the actual air radiation level or radiation dose of crops using a radiation measuring instrument lent by the local government office.”

In addition, we included a roleplay scenario of an example communication situation with a resident who was concerned about the radiation dose level in crops, where the counselor told the resident “There is no problem because the national government has told us that the current benchmark of the dose level of the crops is OK.” After that, we showed a roleplay scenario using a more appropriate form of communication,
“Certainly, some people still have concern about radiation exposure and the dose level, and the perception risk differs among different people. If the problem is just the radiation dose of crops, it will be measured at the local government office, and you can consult with radiation counselor.”

The more specific content of the developed gatekeeper training program is shown in Table 3. To summarize, the differences between the standard MHFA program and this developed program were 1) addressing not only general anxiety but also anxiety for radiation health effects, 2) addressing disaster-related stress and psychological distress (including alcohol and suicide) caused by disaster experience and changes in living environment due to evacuation, and 3) shortening of training time while considering greater accessibility.

### 3.3. Evaluation of the Gatekeeper Training Program for Radiation Health Anxiety and Mental Health Issues

Twelve livelihood support counselors from the Iitate Village Social Welfare Council were trained on 20 November 2019, and 14 counselors from the Minami-Soma Social Welfare Council were trained on 19 December, 2019. The results of the survey of the gatekeeper training program needs among livelihood support counselors showed that most of the counselors of the Social Welfare Council located in the non-evacuation area did not have experience with coping with a resident with radiation health anxiety. Therefore, the intervention subjects were limited to the counselors of Council of Social Welfare located in the evacuation area. All counselors returned the three questionnaires. The average serving periods as a counselor was more than four years, and more than half of the subjects were certificated health or welfare license holders (Table 4).

Comparing the pre-intervention and immediate post-intervention surveys, the degree of self-confidence in counseling and responding to residents increased significantly in all measurements except for one measurement which was “Can you refer a resident with strong anxiety to professionals who can cope appropriately?”, which showed the highest degree of confidence in the pre-intervention period. Among all measurements, “Can you communicate coping skills to a resident with radiation health anxiety?” showed the largest increase in the score (2.46 in pre-intervention to 3.42 in the immediate post-intervention score).

Next, we compared the degree of self-confidence between the pre-intervention survey and the follow-up survey 2-months after the intervention. Compared with the pre-intervention, the confidence level in all measurements was higher than at the baseline, especially, for “How confident are you in supporting a resident with mental health issues?,” which showed the largest increase in the score (1.96 in the pre-intervention period to 3.15 in the follow-up study after 2 months). Regarding the measurements related to the radiation health anxiety, “Can you communicate coping skills to a resident with radiation health anxiety?” (2.46 in the pre-intervention period and 3.15 in the follow-up study; *p* = 0.007) and “Can you refer to a resident with radiation health anxiety to professionals who can cope appropriately?” (3.04 in the pre-intervention period and 3.58 in the follow-up study; p = 0.016) showed significant increases compared with the pre-intervention baseline (Table 5).

## 4. Discussion

This is the first gatekeeper training program developed for individuals who develop radiation health anxiety as well as other mental health issues after a nuclear disaster. Our findings showed a significant increase in coping skills and referral of residents with radiation health anxiety compared to the pre-intervention baseline. Although few studies have measured changes in confidence for each module, such as anxiety or alcoholism, a similar tendency was observed compared with those of previous studies conducted in Japan regarding the confidence of coping with people with depression or suicidal thoughts [19,20,21,22,29].

The results of the survey of training program needs among livelihood support counselors, highlighted the demand not only for listening non-judgmentally but also the need for specific coping skills to support those with radiation health anxiety and other mental health issues such as alcoholism, depression, and suicidal thoughts. Therefore, the gatekeeper training program was focused on “giving reassurance and information” and “encouraging a person to get appropriate professional help,” as well as listening non-judgmentally. Through the gatekeeper training program, the degree of confidence for listening non-judgmentally increased significantly on each category of anxiety (+0.50), radiation health anxiety (+0.40), depression (+0.50), and suicidal thoughts (+0.74) compared with the pre-intervention values. The degree of confidence for referring a resident to appropriate professionals showed a greater increase than listening non-judgmentally for each category (radiation health anxiety (+0.69), alcoholism (+0.92), depression (+0.85), and suicidal thoughts (+0.93)), except anxiety (+0.27), showing the changes in the degree of confidence to the immediately post-intervention from pre-intervention). These findings demonstrate that this training program could help counselors to understand specific ways to cope with residents with radiation anxiety and other mental health issues. In comparison with the pre- and immediate post-intervention values, the skills for communicating radiation health anxiety showed the largest increase in score (2.46 pre-intervention to 3.42 in the immediate post-intervention score). In this gatekeeper training program, we did not spend as much time explaining coping skills related to radiation anxiety as much as other mental health issues such as alcoholism, depression, and suicidal thoughts. In addition, self-confidence in the follow-up survey 2 months after the intervention varied less from the degree of confidence in the immediate post-intervention period. Some items increased from the immediate training period. Therefore, our developed training program could indicate the efficacy at a certain level.

According to NERIS (European Platform on Preparedness for Nuclear and Radiological Emergency Response and Recovery), “the Fukushima accident has highlighted some key issues for further consideration in NERIS research activities, including the key role of access to environmental monitoring and the importance of dealing with uncertainties in assessment and management of the different phases of the accident” [30]. Furthermore, in the evacuation area after the nuclear disaster in Fukushima, there were several opportunities for residents to access health consultations and participate in dialogues about radiation exposure and health concerns [31,32,33]. As in previous reports, we only included specific coping with a resident with a concern about exposure radiation or radiation health anxiety. Therefore, even providing such a simple explanation may lead to a higher degree of confidence among counselors, because the coping skills for radiation health anxiety showed the largest increase in our findings, as well as in a previous report regarding measurement of the radiation dose level while using a risk communication tool [34].

Moreover, the findings of the follow-up survey 2 months after the intervention showed that all measurements improved compared to those at the pre-intervention baseline, and the degree of confidence in communicating with a resident with mental health issues, including radiation health anxiety, increased. Regarding radiation health anxiety, alcoholism, depression, and suicidal thoughts, the degree of confidence in specific coping skills and referring individuals to appropriate professionals showed a significant improvement from baseline. From these findings, communicating and referral skills among the counselors might have been maintained and built upon to a certain degree.

### Limitations and Strengths

The present study has some limitations. First, our intervention design was a single-arm trial study and did not have a control group. Therefore, it may have been influenced by Hawthorne effects [35]. Additionally, the study sample size was small and the statistical power was not sufficient. Besides, not only this gatekeeper training but also other factors, such as on-the-job training, could improve the degree of counselors’ confidence. In summarizing, small sample size without a control group was a severe limitation in this intervention pilot trial study. Therefore, to validate the effectiveness of our newly developed program, a randomized controlled trial with a significant number of subjects is required in future research. The second limitation was that the evaluation of the developed gatekeeper training program was measured by only non-validated subjective items, and there was no objective measurement, such as increasing the number of cases that the counselors referred to appropriate professionals. The third limitation was the lack of evaluation from the residents who consulted with the counselors who took this training program. Fourth, the subjects of the gatekeeper training program were limited to livelihood support counselors of the Social Welfare Council. Besides, this training program was developed based on the needs of livelihood support counselors; therefore, it is necessary to evaluate the general effectiveness of the program. Finally, the authors (Orui, Horikoshi, and Fukasawa) served as instructors in this training program. Therefore, it is necessary to prepare not only the tools of presentation and roleplay but also a feasibility scale for further distribution of the program among supporters who have continued working to reduce radiation health anxiety among residents.

Despite these limitations such as lacking some methods as a stand-alone study, this study has several strengths. Our gatekeeper training program for radiation health anxiety was first developed in Japan and is based on the stated needs for the gatekeeper training program. This developed training program had a positive effect on the communication of counselors with residents with radiation health anxiety as well as other mental health issues.

## 5. Conclusions

We developed a gatekeeper training program for counselors to improve their ability to cope with residents with radiation health anxiety and other mental health issues such as alcoholism, depression, and suicidal thoughts. The degree of confidence in communicating with residents with radiation health anxiety increased compared with pre- and immediate post-intervention surveys. Our findings in the follow-up survey 2 months after the intervention showed significant improvement from baseline in the degree of confidence in using specific coping strategies and referral to appropriate professionals, indicating that the counselors’ skills might have been maintained and built upon to a certain degree. The gatekeeper training program and present study findings will support the ongoing mental health care activities following the Fukushima Daiichi Nuclear disaster and future nuclear disasters that have the potential to occur. For further study, a more generalized and validated gatekeeper training program for radiation health anxiety will be required.

## Figures and Tables

**Figure 1 ijerph-17-04594-f001:**
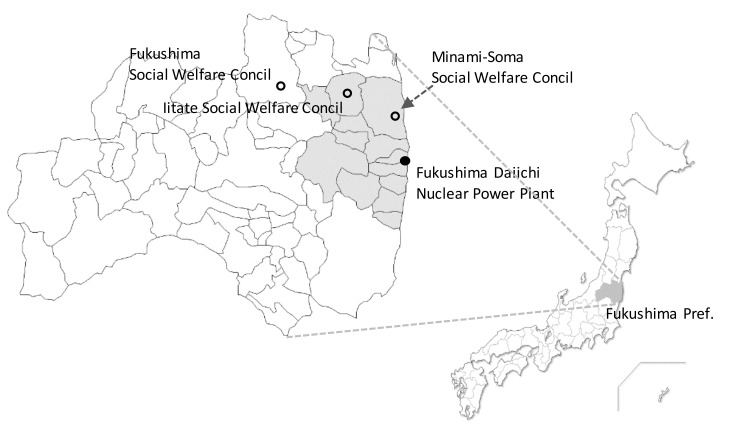
Locations of the Iitate, Minami-Soma, and Fukushima Social Welfare Council and Evacuation areas of the Fukushima Daiichi Power Plant accident: The Social Welfare Councils of Iitate, Minami-Soma, and Fukushima are identified by white circles. The location of the Fukushima Daiichi Nuclear Power Plant shown by a black circle. Regions colored in dark gray correspond to municipalities where evacuation orders were issued.

**Table 1 ijerph-17-04594-t001:** Basic demographics of the subjects who participated in the survey of gatekeeper training program needs among livelihood support counselors.

Basic Demographics	Evacuation/ Non-evacuation				*p* Value
	Evacuation Area (Iitate, Minami-Soma)		Non-Evacuation (Fukushima)		
	(*n* = 30)		(*n* = 10)		
	*n*	(%)	*n*	(%)	
Gender (female)	20	(71.4)	10	(100.0)	0.235
Age					
Less than 40	11	(37.9)	6	(60.0)	
40–59 ages	3	(10.3)	3	(30.0)	
60 age or older	15	(51.7)	1	(10.0)	0.058
Serving periods as a counselor (mean ± SD, years)	4.3 ± 2.5 years		2.1 ± 1.3 years		**0.023 ***
Certificated health or welfare license (nurse, social office worker, etc.)	14	(46.7)	6	(60.0)	0.465
History of coping with condition					
Radiation health anxiety	18	(60.0)	2	(20.0)	**0.029 ***
Other mental health issues	24	(80.0)	3	(30.0)	**<0.001 ***
(Depression)	15	(50.0)	1	(10.0)	**0.025 ***
(Suicide)	6	(20.0)	1	(10.0)	0.471
(Schizophrenia)	9	(30.0)	0	(0.0)	**0.049 ***
(Alcoholism)	12	(40.0)	0	(0.0)	**0.017 ***
(Dementia)	16	(53.3)	2	(20.0)	0.067
(Personality disorders)	2	(6.7)	0	(0.0)	0.402
(Developmental disorders)	1	(3.3)	1	(10.0)	0.402

*Bold: Chi-square test/ t-test p < 0.05, SD: Standard Deviation.

**Table 2 ijerph-17-04594-t002:** Main answers from the survey results of gatekeeper training program needs among livelihood support counselors.

**Radiation health anxiety**
1. Radiation
A counselor answered, “Some residents are concerned about the radiation dose in crops for a while. So, I tell them, if you are worried, you can take it to a place where the radiation dose level of crops can be inspected, and I introduce them to a specialist consultation agency on radiation.”
A counselor answered, “Residents are worried about growing agricultural products.”
Residents said that, “The elderly generation would like to return their hometown, but the young generation do not want to return because they are worried that radiation exposure to children may affect their bodies in the future. As a result, their original family has been separated.”
2. Anxiety
Residents said that, “We are the first example of a nuclear accident in Japan, so we may be used as material for experiments.”
A resident said, “I have anxiety about my health in the future, especially cancer. I am worried about cancer in the future.”
3. Listening non-judgmentally
A counselor answered, “Residents were dissatisfied with TEPCO (Tokyo Electric Power Company) and the national government, so first of all, I always listen slowly, non-judgmentally for a long time, just calming their anger. Listening continuously to their dissatisfaction subsequently builds a relationship”.
A counselor answered, “I am frustrated because I could not give effective advice to residents with concerns about radiation only by listening non-judgmentally.”
**Other mental health issues**
1. Coordination and connection to other specialized facilities
“As a counselor, we listen carefully to mental health issues among residents and continue to follow them in cooperation with public health nurses.”
A counselor answered, “Some evacuees living in the temporary housing were worried about the noise. They wondered about making sounds loudly and disturbing the neighbors. Consequently, they were depressed because of too much worry. So, I referred them to a public health center.”
2. Alcohol or alcoholism
A counselor answered, “One person who lived alone in temporary housing became addicted to alcohol, and he caused many troubles with the neighborhood. However, he was able to get away from alcohol because of being provided with significant support while cooperating with public health nurses and counselors in a mental care center.”
A counselor answered, “I worried about some alcoholics who have a diet that is insufficient in dairy. I just try to keep an eye on them. “I had difficulty coping with a resident with alcohol issues, and my coping skills were insufficient.”
3. Listening non-judgmentally
A counselor answered, “I was sometimes confused about what to say to residents with suicidal thoughts. I would like to learn how to listen non-judgmentally and speak heartfeltly to them.
**Needs for gatekeeper training**
Some counselors answered, “How to cope with residents with alcoholism and suicidal thoughts and obtain more advanced coping skills than listening non-judgmentally.”
A counselor answered, “How to support residents with mental illness while understanding the illness. How to cooperate with other expert facilities (not only “referring” but also communicating with residents with mental illness as a counselor).”
“Training that incorporates roleplay so that counselors can immediately demonstrate what they have learned.”

**Table 3 ijerph-17-04594-t003:** Developed gatekeeper training program contents for radiation health anxiety and other mental health issues.

**A. Presentation and group-work session (70 min)**
1. Introduction: Anxiety for radiation health effects and mental health issues among residents, while having some trouble with family and community relationships or economic issues and being forced to make a decision to return home.
2. Specific coping skills for anxiety related to radiation health effects and other mental health issues
2.1. Depression
2.1.1. Epidemiology of depression and psychological distress among evacuees in Fukushima.
2.1.2. Encouraging early intervention for depressive states.
2.1.3. Five-step principles of the Mental Health First Aid program (MHFA). a) assess the risk of severity of depression and suicide risk, b) listen non-judgmentally, c) give reassurance and information, d) encourage a person to get appropriate professional help, e) encourage self-help strategies.
2.2. Suicide
2.2.1. Epidemiology of suicide in the evacuating area in Fukushima.
2.2.2. Risk and protective factors of suicide.
2.2.3. Specific communicating for residents who have suicidal thoughts. a) ask about the suicidal thoughts, b) encourage a person to get appropriate professional help,
2.3. Alcoholism
2.3.1. Epidemiology of alcoholism and the outline of issues regarding alcoholism.
2.3.2. The difference between heavy drinking and alcoholism.
2.3.3. The psychological factors of the starting to drink alcohol after the Great East Japan earthquake.
2.3.4. Five-step principles of the MHFA. a) assess the risk of alcoholism, b) listen non-judgmentally (encourage to provide helps to residents when they are not drunk), c) give reassurance and information, d) encourage a person to get appropriate professional help, e) encourage self-help strategies.
2.4. Anxiety
2.4.1. An outline of the issues with anxiety.
2.4.2. Group-work (10 min): Let us list the anxiety symptoms (in four dimensions: physical, psychological, behavioral and thought dimensions)
3. Key points of coping with anxiety about radiation health effects and other mental health issues
3.1. Skills for listening non-judgmentally.
3.2. Understanding the trans-theoretical model (stages of change) and ambivalence state of suicidal thoughts and alcoholism among residents.
3.3. Association among depression, alcoholism, and anxiety.
3.4. Specific coping skills for residents who have any anxiety.
3.5. Specific coping skills for anxiety about the health effects of radiation (recommendation to measure the air radiation level or radiation dose of crops).
3.6. Referral or signposting to an appropriate resource or professional.
3.7. Self-help and self-care.
**B. Roleplay session (30 min)**
1. Explanation of the scene-setting: (A 60-year-old woman lived in the ex-evacuation area. She and her husband have returned to the ex-evacuation area. However, her son’s family did not choose to return home because of anxiety regarding the health effects of radiation. Besides, her husband has a significant issue with alcoholism; therefore, she implied having some slight suicidal thoughts to a livelihood support counselor.)
2. Roleplay: (an example of communication for evacuees) and group discussion
Roleplay: (an improved of communication for evacuees) and group discussion

**Table 4 ijerph-17-04594-t004:** Basic demographics of the subjects of the gatekeeper training program for radiation health anxiety and mental health issues.

Basic Demographics	Livelihood Support Counselors in Iitate and the Minami-Soma Social Welfare Council	
	(*n* = 26)	
	*n*	(%)
Gender (female)	19	(65.5)
Age		
Less than 40	9	(36.0)
50s	4	(16.0)
60s and older	10	(40.0)
Serving period as a counselor (mean ± SD, years)	4.4 ± 2.5 years	
Certificated health or welfare license (nurse, social office worker etc.)	14	(53.8)

**Table 5 ijerph-17-04594-t005:** Comparison of the degree of self-confidence among the pre-intervention, immediate post-intervention, and 2-month follow-up surveys.

The Degree of Self-Confidence in Counseling and Responding to Residents	Pre-Intervention		Post-Intervention		Pre/Post *p*-Value	Follow-Up after 2 Months		Pre/2 Months after *p*-Value
	Mean	(SD)	Mean	(SD)		Mean	SD	
**Anxiety (including Radiation health anxiety)**								
Can you listen non-judgmentally to a resident with any anxiety? (*n* = 26)	3.27	(1.04)	3.77	(0.71)	**0.001 ***	3.73	(0.67)	**0.031 ***
Can you communicate coping skills to a resident with any anxiety? (*n* = 26)	2.73	(0.96)	3.54	(0.71)	**<0.001 ***	3.39	(0.75)	**0.002 ***
Can you refer a resident with strong anxiety to professionals who can communicate appropriately? (*n* = 26)	3.46	(1.21)	3.73	(0.83)	0.090	3.69	(0.93)	0.228
Can you listen non-judgmentally to a resident with radiation health anxiety? (*n* = 25)	3.16	(0.94)	3.56	(0.77)	**0.005 ***	3.44	(0.96)	0.110
Can you communicate with a resident with radiation health anxiety? (*n* = 26)	2.46	(1.07)	3.42	(0.76)	**<0.001 ***	3.15	(1.05)	**0.007 ***
Can you refer a resident with radiation health anxiety to professionals who can cope appropriately? (*n* = 26)	3.04	(1.37)	3.73	(0.72)	**0.002 ***	3.58	(1.07)	**0.016 ***
**Alcoholism, Alcohol use disorder**								
Can you explain the psychological background of those who are alcoholic or have an alcohol use disorder? (*n* = 26)	2.23	(0.99)	3.15	(0.83)	**<0.001 ***	3.23	(0.77)	**<0.001 ***
Can you explain appropriate ways of coping with a person who is alcoholic or has an alcohol use disorder? (*n* = 26)	2.15	(1.05)	3.08	(0.84)	**<0.001 ***	2.85	(1.08)	**0.008 ***
Can you refer a resident who is alcoholic or has an alcohol use disorder to professionals who can cope appropriately? (*n* = 26)	2.73	(1.28)	3.65	(0.85)	**<0.001 ***	3.54	(1.03)	**<0.001 ***
**Depression, Suicide**								
Do you have basic knowledge about depression? (*n* = 25)	2.32	(1.11)	3.04	(1.02)	**0.002 ***	3.32	(0.90)	**<0.001 ***
Can you listen non-judgmentally to a resident with depression? (*n* = 26)	2.77	(1.14)	3.27	(1.04)	**0.045 ***	3.42	(0.95)	**0.002 ***
Can you explain appropriate ways of coping to a resident with depression? (*n* = 26)	2.08	(0.93)	2.81	(1.06)	**<0.001 ***	3.08	(0.85)	**<0.001 ***
Do you know who is at risk of suicide? (*n* = 26)	2.00	(0.98)	3.00	(0.98)	**<0.001 ***	2.89	(0.99)	**<0.001 ***
Can you listen non-judgmentally to a resident with suicidal thoughts? (*n* = 26)	2.38	(1.13)	3.12	(1.07)	**0.002 ***	2.96	(1.08)	**0.003 ***
Can you ask calmly about their suicidal thoughts when a resident is considering suicide? (*n* = 26)	2.24	(0.97)	2.88	(0.93)	**<0.001 ***	2.68	(0.99)	**0.005 ***
Can you refer a resident with depression to professionals who can cope appropriately? (*n* = 26)	2.73	(1.28)	3.58	(0.86)	**<0.001***	3.39	(1.13)	**0.002 ***
Can you refer a resident with suicidal thoughts to professionals who can cope appropriately? (*n* = 26)	2.65	(1.29)	3.58	(0.86)	**<0.001 ***	3.27	(1.15)	**0.007 ***
**Confidence for support residents**								
How confident are you in supporting a resident with mental health issues? (*n* = 26)	1.96	(0.87)	2.92	(0.84)	**<0.001 ***	3.15	(0.78)	**<0.001 ***
**Self-care**								
Do you undergo any self-care when you have been stressed? (*n* = 26)	3.08	(0.93)	3.58	(0.70)	**0.007 ***	3.46	(0.95)	0.096
Can you practice your own self-care techniques when you have been stressed? (*n* = 26)	3.12	(0.86)	3.54	(0.95)	**0.013 ***	3.50	(1.03)	0.106
Can you ask for your superiors and colleagues for support when you have been stressed? (*n* = 26)	3.23	(1.18)	3.62	(0.98)	**0.015 ***	3.27	(1.04)	0.814

* Bold: Paired t-test p<0.05; SD: Standard Deviation.

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
