# Peer review of "Development and Evaluation of a Gatekeeper Training Program Regarding Anxiety about Radiation Health Effects Following a Nuclear Power Plant Accident: A Single-Arm Intervention Pilot Trial"

_ijerph, 2020, doi:10.3390/ijerph17124594_

Round 1
Reviewer 1 Report
The study describes the development and a first test of a gatekeeper training programme in the aftermath of the Great East Japan Earthquake. The mental health impact of this major disaster is well-studied. The current study has a different focus, namely the reality that people affected by disasters not always find their way to suitable health services. The authors refer to literature stating the importance of gatekeeper programmes as “prevention strategies aimed at educating social and community facilitators to identify signs of suicidal behaviour and to signpost individuals to appropriate services”. This fits well within post-disaster psychosocial support guidelines (and reviews of psychosocial interventions in a CBRN context) that are available, recommending early detection of mental health and psychological problems combined with access to professional health care where appropriate. They illustrate how the programme was developed based on input from the target-group of participants and assessed the self-confidence of participants “when they counselled with residents who have had radiation health anxiety and other mental health issues” at two moments in time. The findings suggest that the counsellors feel better equipped to perform in the role of gatekeepers. I think the paper is informative to policy makers and scholars who seek to strengthen the evidence base of capacity building in the field of mental health and psychosocial support. However, I identified some issues I am confident the authors are able to address to further improve the manuscript.
#1. Objective in the introduction. Line 85: “we evaluated the efficacy of the developed program”. This is misleading as it suggests that the authors assessed whether the counsellors enrolled in the gatekeeper training contributed to better mental health within communities. In fact, the impact on mental health symptoms or disorders is not examined. The authors should modify the text to raise the right expectations.
#2. It is logical to ask the target group of end-users about their background, experiences, and what they want to gain/expect from a training (as is worked out in para 2.1). Nevertheless, given that quite some studies have been published on gatekeeper programs (especially in relation to suicide) the authors should say more about elements they considered relevant and why. In the end, it seems a bit odd to refer to international research to emphasize the importance of a training programme and to ignore the international knowledge base during the development of the Japanese training.
#3. In addition to the previous point: Instead, without further introduction we read (line 115) that an existing programme is used as a starting point: the “Mental Health First Aid program (MHFA), which has already been developed and proven to be effective [23–25]”. Two questions: (1) What is the difference between MHFA and the new gatekeeper programme? (2) How was the effectiveness of MHFA determined?
#4. Lines 185-189: “The standard MHFA program is a 12-h course applicable to diverse conditions including depression, anxiety disorders, psychosis, and substance use disorder. However, we set this gatekeeper training program to a 100-minute course because it took a consideration into more accessible and less burden program when counsellors take the training.” Which elements were left out in the eightfold reduction of the original programme? Based on which rationale did the authors determine to crop a programme that was evaluated as effective? Does this mean the original programme contains redundant modules, or in other words, is less cost-effective as it could be?
#5. Line 210: Only the part on “radiation health anxiety and other mental health issues” is included in the manuscript. Is it possible to include an English and a Japanese version of the entire programme as an online Supplement (this is also linked to a recent conclusion of a meta-analysis of psychological intervention manuals that original manuals or often not available; Watts et al. 2020).
#6. Table 5: Apart from improved scores after the training, in virtually each item the variation decreased shortly after the training (and decreased further in the items that further improved in the follow-op), I would emphasize this as a potential effect of the training as well in addition to the conclusions concerning increased self-confidence - less variation among participants post-training compared to pre-training, in combination with higher scores on items, is also indicative for training efficacy.
#7. With #2 in mind, I want to encourage the authors to reflect more in the discussion on similarities and differences of this programme compared to other gatekeeper training programmes. Can they recommend elements that might be relevant for programmes in other countries or event types?
#8. Counsellors better understand “specific ways to cope with residents with radiation anxiety and other mental health issues”. This is a promising first step but modesty is required as the study is conducted in a limited sample without a control group. The authors should make reference to placebo or Hawthorne effects and recommend thorough replications in stronger controlled study designs with comparison groups.
#9. Lines 277-283: Last but not least, you can present this subsection as a positive message, but it is left implicit through which mechanism the training contributes to disaster mental health. Currently we have no indication of a positive effect on the actual health service delivery/quality of care. A replication with a stronger design (multiple arms, larger samples) alone (see #8) is not sufficient; counsellors are one side of the story, the people they help should be included in future research as well, with a control group.
Minor point:
Line 78. Change ‘“the Basic (…)’ into ‘the “Basic (…)’
Author Response
Thank you for your polite reviewing and suggesting important comments. As you pointed out, we have revised our manuscript.
I really appreciate for your works.

Reviewer 2 Report
Comments for authors from reviewer
This manuscript is about an important topic, and I appreciate the intervention emphasis. However, I raise some concerns below that must be addressed before publication could be recommended.
Authors state: [55-57] “Residents in the Fukushima evacuation area not only have concerns about exposure radiation 55 and the effects on their health, but also other mental health problems (overdrinking alcohol or 56 alcoholism, depression, and suicide) related to the nuclear disaster and subsequent changes in their 57 personal lives and environment due to the evacuation [8–13].” This should be elaborated on further as it is part of the rationale for this specific intervention.
This same para mentioned above goes directly into an explanation of the ‘livelihood support counselors’. I suggest this should be a separate paragraph and the rationale underlying the deployment of these counselors should be explained.
In the following paragraph [65] more information should be provided about the gatekeeper model. Does previous research/evaluation support this model in terms of reduction in risk of suicide or other outcomes?
The introduction includes [69-73] a description of the current study. I suggest this be moved to the end of this intro section (perhaps under a ‘current study’ subheading). Furthermore, this introductory section concludes with a mention of the findings. This information should be saved for the results and discussion sections of the manuscript.
The Methods and Materials section is sparse and needs more detailed information about sampling, participant selection, measures, procedures and more. I believe a total of 40 counselors were administered initial questionnaires [line 143], although this is somewhat unclear based on other references in the text and tables.
A reference to data analysis is also included in this section [100-102] but is lacking in necessary detail. In addition, more information on the process of developing the intervention is required. It is not clear how this intervention differs from MHFA, how it was implemented, duration of intervention and more. However, some additional details on this are included in the Results section [starting line 183 and continuing throughout]. This information should be moved to Methods.
It seems this is a pre-post design and an additional follow up time point 2 months out – but no comparison group. Only 26 counselors seem to have participated in the intervention. This is a weak design with a small sample size, calling into question results and the subsequent interpretation.
The rationale and implications of collecting responses from counselors in both evacuation and non-evacuation zones is unclear. In addition, it is confusing at times to understand how responses are framed in terms of counselor compared to resident perspectives.
I do appreciate the various concerns raised in the Limitations section of the manuscript, but I don’t believe these limitations have been adequately addressed to justify publication. It might be more appropriate if this were framed as a pilot or exploratory study, Step 1 in a longer and more rigorous process.
Author Response

(The authors gave the same response as above.)

Round 2
Reviewer 1 Report
I believe the authors responded well to my comments and the feedback from the other reviewer. I have no additional points to make.
Author Response
Thank you for your reviewing and comments.

Reviewer 2 Report
Thank you, I believe you have adequately addressed comments from first round reviews. This is an interesting study if considered as an exploratory endeavor. The topic is important and the manuscript is compelling if considered as Step 1 in a larger study, ideally with a true control comparison and larger sample size. I think it is important to more clearly acknowledge significant limitations related to the research design (lack of control, small sample size) at the start of the manuscript. If you hope to do another more rigorous study as Step 2, expanding on these efforts, please mention this at the outset. As a stand alone study the methods are lacking, but if seen in broader context this is a compelling contribution.
Author Response
Thank you for your reviewing and comments. As your suggestion, we have revised our manuscript. Could you please see a attached file.
Thank you.